# A Hydrolase Produced by *Rhodococcus erythropolis* HQ Is Responsible for the Detoxification of Zearalenone

**DOI:** 10.3390/toxins15120688

**Published:** 2023-12-07

**Authors:** Junqiang Hu, Shilong Du, Han Qiu, Yuzhuo Wu, Qing Hong, Gang Wang, Sherif Ramzy Mohamed, Yin-Won Lee, Jianhong Xu

**Affiliations:** 1Key Laboratory of Agricultural Environmental Microbiology, Ministry of Agriculture, College of Life Sciences, Nanjing Agricultural University, Nanjing 210095, China; 2021216027@stu.njau.edu.cn (J.H.); 2020816121@stu.njau.edu.cn (S.D.); mollyqh@163.com (H.Q.); 2Jiangsu Key Laboratory for Food Quality and Safety-State Key Laboratory Cultivation Base, Ministry of Science and Technology/Key Laboratory for Agro-Product Safety Risk Evaluation (Nanjing), Ministry of Agriculture and Rural Affairs/Key Laboratory for Control Technology and Standard for Agro-Product Safety and Quality, Ministry of Agriculture and Rural Affairs/Collaborative Innovation Center for Modern Grain Circulation and Safety/Institute of Food Safety and Nutrition, Jiangsu Academy of Agricultural Sciences, Nanjing 210014, China; njcpuwg@126.com (G.W.); lee2443@snu.ac.kr (Y.-W.L.); 3School of Food and Biological Engineering, Jiangsu University, Zhenjiang 212013, China; 2222218095@stmail.ujs.edu.cn; 4Food Industries and Nutrition Research Institute, Food Toxicology and Contaminants Department, National Research Centre, Tahreer St., Dokki, Giza 12411, Egypt; sheriframzy4@gmail.com; 5Department of Agricultural Biotechnology, Seoul National University, Seoul 08826, Republic of Korea

**Keywords:** zearalenone, *Rhodococcus erythropolis*, biodegradation, hydrolase

## Abstract

Zearalenone (ZEN), an estrogenic mycotoxin, is one of the prevalent contaminants found in food and feed, posing risks to human and animal health. In this study, we isolated a ZEN-degrading strain from soil and identified it as *Rhodococcus erythropolis* HQ. Analysis of degradation products clarified the mechanism by which *R. erythropolis* HQ degrades ZEN. The gene *zenR* responsible for degrading ZEN was identified from strain HQ, in which *zenR* is the key gene for *R. erythropolis* HQ to degrade ZEN, and its expression product is a hydrolase named ZenR. ZenR shared 58% sequence identity with the hydrolase ZenH from *Aeromicrobium* sp. HA, but their enzymatic properties were significantly different. ZenR exhibited maximal enzymatic activity at pH 8.0–9.0 and 55 °C, with a Michaelis constant of 21.14 μM, and its enzymatic activity is 2.8 times that of ZenH. The catalytic triad was identified as S132-D157-H307 via molecular docking and site-directed mutagenesis. Furthermore, the fermentation broth of recombinant *Bacillus* containing ZenR can be effectively applied to liquefied corn samples, with the residual amount of ZEN decreased to 0.21 μg/g, resulting in a remarkable ZEN removal rate of 93%. Thus, ZenR may serve as a new template for the modification of ZEN hydrolases and a new resource for the industrial application of biological detoxification. Consequently, ZenR could potentially be regarded as a novel blueprint for modifying ZEN hydrolases and as a fresh resource for the industrial implementation of biological detoxification.

## 1. Introduction

Zearalenone (ZEN) is an estrogenic mycotoxin produced by several *Fusarium* species during their secondary metabolism process. It is commonly present in cereal crops such as wheat and corn, as well as their products [1,2]. ZEN is a structural analog of estradiol, which can bind to estrogen receptors, leading to reproductive disorders in animals. A high concentration of ZEN can even induce apoptosis [3]. In order to mitigate the losses caused by ZEN contamination in the feed and livestock industries, physical and chemical methods of ZEN reduction have been employed [4]. However, these techniques have limitations. Physical sorbents have poor specificity and absorb ZEN and nutrients indiscriminately [5,6], whereas chemical reagents are both inefficient and potentially leave behind reagent residues in the treated material [4,7]. Biological detoxification, on the other hand, has the benefits of safety and high specificity, and can disrupt the structure of ZEN to accomplish total detoxification [8,9].

To date, peroxidase [10], laccase [11,12], and hydrolase are the three types of enzymes known to disrupt the structure of ZEN [13,14]. Peroxidase degrades ZEN with the assistance of H_2_O_2_ [15], whereas laccase requires the addition of exogenous copper ions to assist in the breakdown of ZEN [16]. ZEN hydrolases have been extensively researched due to their well-defined degradation mechanism which does not require cofactor addition during the process [17]. Since the ZEN-degrading enzyme Zhd101 was isolated from *Clonostachys rosea* [13], ZEN hydrolases with high amino acid sequence similarity to Zhd101, including RmZHD, CbZHD, and ZENG have been reported [14,18,19]. These enzymes exhibit poor thermal stability and low enzyme activity under acidic conditions in practical applications. As a result, researchers have initiated efforts to enhance their performance through modifications [20]. Due to the high degree of similarity among the reported ZEN-degrading enzymes, a single species of enzyme with extremely similar enzyme properties has evolved [17,21]. Due to the above, there are insufficient enzyme templates available for a rational design. To address this limitation, there is a need to screen new ZEN-degrading bacteria from the environment and explore new types of ZEN-degrading enzymes.

In this study, we screened for a ZEN-degrading strain, identified as *Rhodococcus erythropolis*, and elucidated the mechanism of ZEN degradation by *Rhodococcus* sp. for the first time. On this basis, we identified a ZEN-degrading enzyme named ZenR from *R. erythropolis* and characterized the kinetic properties and essential amino acid residues of ZenR. Additionally, we achieved successful expression of ZenR as a secreted protein in *Bacillus subtilis*. This study expands the pool of strains and enzymes that are able to degrade ZEN, thereby establishing a foundation for the future application of ZEN-degrading enzyme preparations.

## 2. Results and Discussion

### 2.1. Isolation, Identification and Characteristics of the ZEN Degradation Strain

An aerobic bacterium isolated from soil (Suzhou city, Anhui province), which displayed a high ZEN-degrading activity, and was named HQ. Following a 3 d incubation period at 30 °C on LB agar, the colonies of the HQ strain were observed to be circular, convex, with smooth margins, opaque, and pale orange in color (Appendix A). Further analysis revealed that strain HQ was Gram-positive bacterium with a rod-shaped morphology (Appendix A). In physiological and biochemical analyses, strain HQ tested positive for urease but negative for lipase. Strain HQ utilized pyruvate, D-maltose, and D-malate as the sole carbon sources and arginine and L-glutamine as the sole carbon and nitrogen sources (Appendix A), consistent with *Rhodococcus erythropolis* djl-11 [22]. The 1.5 kb 16S rRNA fragment of strain HQ is 99% identical to *Rhodococcus erythropolis* NBRC 15567 (BCRM01000055). Strain HQ was identified as *Rhodococcus erythropolis* through morphology observation, physiological and biochemical characteristics, and 16S rRNA and phylogenetic analysis (Appendix A). The 16S rRNA sequence of strain HQ has been deposited in the GenBank database as Accession No. OR497749.

The optimal growth temperature for *R*. *erythropolis* HQ in LB medium was 30 °C. Strain HQ grew slowly at 35 °C and ceased growing at 40 °C. The optimal growth pH range for growth of strain HQ was between 6.0 and 8.0, and growth terminated at the initial pH of 4.0 and 10.0 (Appendix A). The degradation rate of ZEN was significantly affected by temperature and pH. Its optimal degradation temperature was 30 °C and its optimal degradation pH value was 9.0. Under optimal degradation conditions, strain HQ was able to degrade 95% of the ZEN within 48 h in MSM medium with an initial ZEN concentration of 15 μg/mL. Notably, there was a 24 h delay before the strain degraded the ZEN, which could be due to a low inoculum dose or lack of fast-acting carbon sources in MSM required for strain HQ growth.

Due to the exceptional aromatic compound degradation capabilities of *Rhodococcus* sp. [23], researchers had explored the degradation of ZEN by *Rhodococcus*. Risa et al. evaluated 42 standard strains of *Rhodococcus*, and results showed that only *Rhodococcus percolatus* JCM 10087 exhibited the ability to degrade ZEN [24]. Cserháti et al. examined 32 strains of *Rhodococcus*, discovering that 13 strains of *R. erythropolis* were unable to degrade ZEN, whereas three strains of *R. pyridinivorans* (K402, K404, K408) exhibited high ZEN-degrading activity [25]. Kriszt et al. conducted an immature uterotrophic experiment to assess the ability of *R. pyridinivorans* K408 to mitigate the estrogenic effects of ZEN. The degradation products produced by *R. pyridinivorans* K408 demonstrated no residual estrogenic effects [26]. However, existing studies primarily focused on the degradation rate of ZEN by *Rhodococcus* and the toxicity of the degradation products, lacking in-depth exploration of the metabolic pathways involved or the specific genes participating in ZEN degradation.

### 2.2. Identification of the ZEN Degradation Product of R. erythropolis HQ

To elucidate the chemical mechanism of ZEN degradation by *R. erythropolis* HQ, the degradation product of ZEN was identified. Fermentation broths, collected at different time points, were lyophilized and then reconstituted in acetonitrile for HPLC analysis. A distinct degradation product was detected at a retention time of 10.75 min after 26 h of fermentation, exhibiting a gradual accumulation concomitant with ZEN degradation (Appendix A). Employing a more sensitive LC-TOF-MS/MS system, the chemical formula of the degradation product was determined to be C_18_H_24_O_6_ through analysis of the negative ion ESI-MS data (*m*/*z* = 336.1546). It was hypothesized that the chemical structure of degradation product contained one more molecule of H_2_O than that of ZEN (Figure 1A,B). Further interpretation of its MS/MS spectrum revealed fragment ions corresponding to ZEN moieties. Consequently, it was concluded that ZEN underwent a cleavage of the lactone ring and identified the degradation product as (S, E)-2,4-dihydroxy-6-(10-hydroxy-6-oxoundec-1-en-1-yl) benzoic acid, named P1.

The degradation product, P1, served as a substrate for strain HQ, leading to its complete degradation within 5 d. In the control group without strain HQ, P1 underwent partial spontaneous degradation, resulting in the accumulation of a new product, named P2. Through analysis of the negative ion ESI-MS data (*m*/*z* = 292.1759) and MS/MS spectrum (Figure 1C), the chemical formula of product P2 was determined to be C_17_H_24_O_4_. With a molecular weight 44 mass units lower than P1, it was speculated that P2 originated from the decarboxylation of P1. Therefore, we proposed P2 to be (S, E)-1-(3,5-dihydroxyphenyl)-10-hydroxyundec-1-en-6-one. This hypothesis was confirmed by a comparison of the experimental data with the published literature [27,28].

To assess the chemical stability of P1, experiments were conducted using acetonitrile, water, and methanol as solvents. Notably, P1 exhibited relatively high stability in acetonitrile while undergoing spontaneous degradation in water and methanol (Appendix A). Based on these results, we postulated that P1 would demonstrate stability in an aprotic solvent. Subsequently, the stability of P1 in acetone and ethanol was tested, revealing that P1 exhibited stability exclusively in acetone, an aprotic solvent. This observation suggests that P1 acts as an intermediate during the degradation process. The degradation pathway of ZEN by strain HQ is shown in Figure 1D.

### 2.3. Expression and Identification of Enzyme ZenR

We obtained a 7.09 M draft of the genome containing 80 scaffolds and 6948 annotated genes via high-throughput sequencing. The DNA sequences of *R. erythropolis* HQ chromosomes have been deposited in the NCBI SRA database with BioProject accession number PRJNA1010513. Based on the degradation pathway of ZEN by *R. erythropolis* HQ, it is hypothesized that strain HQ possesses an α/β hydrolase capable of degrading ZEN. To investigate this further, we employed the reported sequences of the ZEN hydrolase as templates. Through a local blastp analysis, we aligned an amino acid sequence that exhibited 58% identity to the template sequence ZenH (Accession No. WP_269305865). Subsequently, the target gene was heterologous expressed in strain BL21 (DE3) and whole-cell catalysis experiment confirmed that the enzyme, named ZenR (Accession No. WP_081559690.1), could hydrolyze ZEN. The phylogenetic tree constructed from the amino acid sequences of ZenR and known ZEN degrading enzymes revealed that ZEN hydrolases are closely related to Zhd101 (Accession No. AB076037), which belongs to the α/β hydrolase_1 family, whereas ZenR belongs to the α/β hydrolase_6 family (Figure 2A). Sequence comparison by the Lipman–Pearson method showed that the amino acid sequence similarity between ZenR and ZENC was only 23% [29,30].

Since the C-terminus of ZenR was fused with a His-tag, it was purified by single-step nickel affinity chromatography, in which 150 mM imidazole eluted the ZenR adsorbed on the nickel column (Appendix A). The results indicated that ZenR was highly expressed in BL21 (DE3), and 8.2 mg of the purified enzyme was obtained from 100 mL of the LB medium. The subunit of ZenR formed a single band on SDS-PAGE (Figure 2B) and its molecular weight of 36.6 kDa was consistent with the theoretical subunit weight. The molecular weight of the recombinant protein was approximately 40 kDa, as determined by native PAGE (Figure 2C), indicating that ZenR was a monomeric form rather than a dimeric or multimeric form.

The *zenR* gene in strain HQ was eliminated using a suicide plasmid, and the ZEN-degrading ability of the mutant ΔzenR-HQ was examined to determine the presence of additional ZEN-degrading genes in strain HQ. As shown in Appendix A, the mutant strain ΔzenR-HQ exhibited a significantly reduced capacity to degrade ZEN in comparison to strain HQ. However, the mutant strain retained the capability to degrade approximately 60% of ZEN within 3 d. The above results indicate the presence of other genes responsible for ZEN degradation besides ZenR in strain HQ, with the *zenR* gene being identified as the key gene responsible for ZEN degradation. This is the first time that a ZEN-degrading gene has been identified in *Rhodococcus* species.

### 2.4. Characterization of Enzyme ZenR

The optimal temperature range for the recombinant enzyme ZenR was 50 °C to 55 °C and ZenR lost the majority of its enzyme activity at 65 °C (Figure 3A). The optimal pH range of ZenR was 8.0–9.0 (Figure 3B), which is similar to those of the most ZEN degrading enzymes, including RmZHD [18], CbZHD [19], and Zhd518 [20]. The specific activity of ZenR towards ZEN was 20.04 U/mg, and its *K_m_*_,*app*_ and *V_max_* values were 21.14 ± 0.16 μM and 0.25 ± 0.012 μmol/s*mg, respectively (Figure 3D).

Although the sequence similarity between ZenR and ZenH (optimum pH 7.0) is as high as 65%, there was a significant difference between the two enzymes in terms of optimal pH. As shown in Appendix A, the majority of the amino acid residues that differed between the two sequences were α-helix and surface residues (Appendix A), whereas the amino acid residues in the protein’s core region were conserved. Previous studies demonstrated that the electrostatic potential could significantly change the optimal pH of the enzyme [31], negatively charged amino acids were introduced on the protein surface to decrease the optimal pH [32,33], while arginines were introduced on the surface of α-amylase to increase the optimal pH [34,35]. Appendix A demonstrate that the electrostatic potential on the surface of enzyme ZenR is significantly greater than that of ZenH, and this may be one of the reasons why ZenR has a significantly higher optimal pH than ZenH. The optimal temperature range for the majority of the reported ZEN hydrolases is in the range of 37–45 °C [14], while the optimal temperature for ZenR is 55 °C. ZenR has the maximum known enzyme activity, which is 2.8 times that of ZenH and 1.3 times that of Zhd101 under optimal conditions.

ZenR activity was inhibited by Fe^3+^, Fe^2+^, Zn^2+^, Ni^2+^, Mn^2+^, Cu^2+^, and Ca^2+^, with Fe^3+^ and Zn^2+^ inhibiting more than 50% of the activity. Approximately 70% enzyme activity was retained by the addition of Ni^2+^, Mn^2+^, Cu^2+^, and Ca^2+^. In addition, Co^2+^, Mg^2+^, and EDTA had no effect on ZenR activity. Therefore, ZenR does not require metal ions as an activator to ensure enzyme activity and the metal ion-independent property is advantageous for use in the feed industry.

The thermostability of enzymes is a very important feature in applications. ZenR retained approximately 80% of the enzyme activity when incubated at 40 °C for 10 min but lost all activity after incubation at 45 °C for 4 min (Figure 3E). These results showed that the thermostability of ZenR is significantly higher than ZenH. ZenH nearly lost its enzyme activity after incubation at 40 °C for 2 min [17]. As shown in Appendix A, the cap domain is composed of 5 α-helices and 2 β-sheets. A structural analysis based on the B-factor revealed that the cap domains of both enzymes were unstable, but the sixth and tenth α-helices in the ZenR cap domain are significantly more stable than the corresponding α-helices in ZenH. The thermostability of enzymes is influenced by a variety of factors, among which salt bridges play a crucial role in the structure of proteins [36,37]. Although both enzymes have 10 pairs of salt bridges, ZenR has more salt bridges in the cap domain than ZenH, as well as a pair of salt bridges between the region with the highest B-factor (between eighth β-sheet and the eighth α-helix), which significantly increases the rigidity of the region, thereby enhancing the stability of the protein. In addition, ZenR could not tolerate incubation at 45 °C in thermostability experiments but its enzyme activity was maintained at 55 °C, most likely due to the formation of a ZEN-enzyme complex, which stabilizes the conformation of the enzyme.

The resistance of ZenR to organic solvents was determined by exposing it to ethanol at various concentrations. The enzyme activity of ZenR gradually decreased with an increase in ethanol concentration. ZenR lost its enzyme activity when the ethanol concentration reached 15%. It is hypothesized that either ethanol denatured the enzyme or the presence of ethanol caused the loss of essential water molecules on the protein’s surface [38], resulting in ZenR losing its ability to degrade ZEN. ZenR can be used to detoxify feed containing minimal alcohol concentrations. Typically, the alcohol concentration in the product of alcoholic fermentation is below 20% [39].

### 2.5. Structure-Based Sequence Analysis

Homology modeling of ZenR was carried out by the ColabFold website and five models were obtained. Model III with the highest score was selected for further experiments and analysis (Appendix A). As shown in Appendix A, ZenR consisted of three components: the cap domain, the substrate catalytic pocket, and the core domain, where the cap domain contains five α-helices and two β-folds, and the core domain consists of eight β-folds and six α-helices surrounding the β-folds. The core domain of ZenR is similar to that of Zhd101, but the cap domain of ZenR has one additional α-helix and two additional β-sheets compared to Zhd101 (Appendix A), which corresponds to the two additional amino acid sequences in ZenR as compared to Zhd101 (Appendix A).

The enzyme active center determines the specific recognition and binding of the enzyme to the substrate. By determining the enzyme active center, enzyme preparations can be developed more efficiently. The catalytic triad, consisting of Ser-Asp/Glu-His, is a prominent feature in α/β-hydrolases and Ser is typically situated in the center of a consensus motif of Gly-X-Ser-X-Gly, where “X” represents any amino acid. Using AutoDock software (version 4.2.6), ZEN was docked to the substrate binding pocket of ZenR and subsequently searched for key amino acid residues near ZEN. The catalytic triad residues were presumed to be Ser132-Asp157-His307 (Figure 4). The Consensus Finder was employed to obtain all sequences with more than 30% similarity to ZenR, and these sequences were then subjected to analysis using Weblogo 3.0 to visualize the conserved regions. The analysis confirmed that the predicted catalytic triad of amino acids is conserved in the sequence of ZenR (Appendix A). To verify the functionality of the predicted catalytic triad, three mutants, S132A, D157A, and H307A, were constructed via site-directed mutagenesis, and the mutation results were confirmed by gene sequencing. All proteins expressed by the three mutants remained soluble, suggesting that the mutation of the active center amino acid to alanine did not induce significant changes in the enzyme structure. This lack of activity was attributed to the disruption of crucial functional residues in the active center caused by the alanine substitution (Appendix A). Consequently, the catalytic triad of ZenR was determined to be Ser132-Asp157-His307, which is identical to that of the enzyme ZenH [17]. Based on our results and a previous study [19,40], it was hypothesized that ZenR utilized the serine hydrolase mechanism. The oxygen on Ser132 functions as a nucleophile to attack the carbonyl C atom of ZEN in order to form a covalently bound intermediate. The C-O bond of the acyl-enzyme intermediate is subsequently cleaved, while the proton transferred to His307 creates a new hydroxyl group with the oxygen atom of ZEN, ultimately resulting in the cleavage of the macrocyclic lactone structure.

### 2.6. Application of Recombinant Bacillus in Cornmeal

The recombinant *Bacillus* fermentation broth was applied to ZEN-contaminated cornmeal and its fermentation products to evaluate the ZEN biodegradation ability of the recombinant enzyme ZenR. After treating the raw cornmeal with the fermentation broth for 3 h, the ZEN content declined from 2.87 μg/g to 1.05 μg/g, with a degradation rate of 63.4% (Figure 5). Furthermore, when the fermentation broth was added to cornmeal liquefied by α-amylase, the ZEN residue in the sample was significantly reduced to 0.21 μg/g, which falls below the regulatory limit of 0.5 μg/g for ZEN in animal feed. In cornmeal, zein can bind to ZEN through non-covalent interactions [41,42], preventing the hidden ZEN from being exposed to ZenR and thus reducing the degradation rate. However, when cornmeal was liquefied under high-temperature conditions, the hidden ZEN with the matrix was completely released [43], leading to a substantial increase in the degradation rate of ZEN by ZenR. In contrast, when the fermentation broth was added to the fermented samples, the ZEN residue remained high at 2.19 μg/g, with a degradation rate of 23.7% (Figure 5). This can be attributed to the significant decrease in pH value after *Saccharomyces cerevisiae* fermentation [44]. The low pH value greatly suppressed the enzymatic activity of ZenR, resulting in the majority of ZEN remaining in the sample. Therefore, the recombinant Bacillus fermentation broth can be effectively applied to liquefied corn samples, significantly reducing the ZEN content of corn raw materials for animal feed.

## 3. Conclusions

We isolated and characterized a ZEN-degrading strain from soil, identified as *Rhodococcus erythropolis*. *R. erythropolis* HQ, specifically, demonstrated the ability to cleave the lactone bond of ZEN. Through local blastp analysis, we identified the ZEN degradation gene, *zenR*, in strain HQ. The results of the gene knockout experiment confirmed *zenR* as the key gene responsible for ZEN degradation in this strain. This study provides the first elucidation of the ZEN degradation mechanism of *Rhodococcus* species. Upon heterologous expression of *zenR*, we observed that the enzyme ZenR displayed maximum activity at 55 °C and pH 8.0–9.0. ZenR is structurally similar to the previously described ZEN-degrading hydrolase, ZenH, but it exhibits superior thermal stability. This increased stability is likely attributed to changes in salt bridges on the protein surface of the two enzymes. Utilizing multiple sequence alignment, molecular docking, and site-directed mutagenesis, we identified the catalytic triad of ZenR as S132-D157-H307, classifying ZenR as a typical α/β hydrolase. Moreover, we successfully expressed ZenR as a secreted protein in *Bacillus subtilis*, laying the groundwork for future research. These findings deepen our comprehension of ZEN degradation and underscore the biotechnological potential of ZenR.

## 4. Materials and Methods

### 4.1. Soil Samples, Chemicals and Reagents

Soil samples were collected from the mycotoxin pollution area of Suzhou City in Anhui Province, China. ZEN (>98%) was diluted in in acetonitrile as a standard stock solution (10 mg/mL), which was obtained from Romer Labs (Beijing, China). The ClonExpress II one step cloning kit was purchased from Vazyme Biotech Co., Ltd. (Nanjing, China). Minimal salt medium (MSM) contained 1.0 g/L KH_2_PO_4_, 1.6 g/L Na_2_HPO_4_, 0.5 g/L NaNO_3_, 0.5 g/L MgSO_4_·7H_2_O, 0.025 g/L CaCl_2_·2H_2_O and 0.5 g/L (NH_4_)_2_SO_4_. Luria-Bertani (LB) medium contained 5 g/L yeast extract, 10 g/L peptone, and 10 g/L NaCl. The bacterial strains and plasmids used in this study are listed in Appendix A. The primers used for PCR amplification of DNA are listed in Appendix A.

### 4.2. Isolation and Identification of Strain HQ

Soil samples (2 g) were dispersed in 20 mL of sterile water and allowed to settle for 10 min, and then 500 μL of the resulting supernatants were incubated in 4.5 mL of MSM supplemented with 50 mg/L ZEN. The cultures were incubated at 30 °C for 7 d with shaking at 180 rpm. Each culture was then transferred to fresh MSM medium containing 50 mg/L ZEN for an additional 7 d under the same conditions, and this procedure was repeated three times. Antibiotics such as ampicillin (100 mg/mL) and kanamycin (50 mg/mL) were administered to ZEN-degrading cultures to reduce microbial populations without influencing ZEN degradation. After the antibiotic treatment, aliquots of the gradient-diluted cultures (0.1 mL) were distributed onto plates containing 10-fold-diluted LB medium and incubated at 30 °C for 5 d. Individual colonies were randomly isolated and evaluated for their degradation activity. Strains capable of degrading ZEN were stored in 30% glycerol at −80 °C until use.

The selected strain HQ was identified by its 16S ribosomal RNA and biochemical tests. The bacterial genomic DNA was extracted using the EZ-10 DNA isolation kit (Sangon Biotech, Shanghai, China) according to the manual’s instructions. The partial 16S rDNA gene was amplified by PCR using the universal primers 27F and 1492R. The PCR product was then inserted into pMD18T vector via the TA cloning kit (TaKaRa Co., Ltd., Dalian, China), the plasmid was subsequently transformed into competent *E. coli* DH5α cells and the positive clone was sequenced by Sangon Biotech (Shanghai, China). The resultant 16S rRNA was analyzed by performing a blast search on the EZBioCloud database (https://www.ezbiocloud.net/ (accessed on 10 January 2023)). In addition, a neighbor-joining (NJ) phylogenetic tree of the 16S rRNA sequence from 17 bacterial species was constructed using the MEGA X program, and evolutionary distances were calculated according to Kimura’s two-parameter model [45]. In this case, bootstrap values were calculated based on 1000 replicates.

Biochemical characterization of strain HQ was performed by Zhejiang Tianke High-Tech Technology Development Co., Ltd. (Hangzhou, China) using the VITEK 2 compact bacterial identification system (bioMérieux, Lyon, France) combined with VITEK 2 identification cards.

### 4.3. Growth and Degradation Characteristics of the Strain HQ

To determine the optimal growth conditions for the strain HQ, a single colony was inoculated into a liquid LB medium to prepare a seed culture. When the OD_600_ value of strain HQ was reached 0.6–0.8, the seed cultures were inoculated at 1% into 100 mL of fresh LB medium (pH 6.8) and incubated at various temperatures (20 °C to 40 °C) and pH values (4.0 to 10.0) with shaking at 180 rpm, respectively. Cultures were collected at regular intervals to determine OD_600_ values. The strain HQ was washed with distilled water and resuspended in PBS buffer (pH 7.0) to assess the degradation of ZEN. The test conditions included an incubation temperature range of 20 °C to 35 °C and a pH range of 5.0 to 10.0. The bacterial suspension was inoculated at a concentration of 1% into 10 mL MSM containing 15 μg/mL ZEN, and incubated with shaking at 180 rpm. ZEN concentrations were quantified using HPLC from samples collected at regular intervals. Each treatment group was executed three times.

In order to explore the ZEN degradation ability of the mutant strain ΔzenR-HQ, both the strain HQ and ΔzenR-HQ were inoculated into MSM medium containing 10 μg/mL ZEN at the same inoculation amount, respectively. The cultures were then incubated at 30 °C and 180 rpm for 3 d.

### 4.4. DNA Sequencing, Assembly, and Annotation

Genomic DNA of high quality was isolated from the cell pellets with a Bacteria DNA Kit (Omega Biotech, Guangzhou, China) according to the manufacturer’s instructions, and quality control was subsequently carried out on the purified DNA samples. Sequencing of the strain HQ draft genome was performed by Shanghai Biozeron Biotechnology Co., Ltd. (Shanghai, China) using the Illumina novaseq 6000 sequencing platform. The raw paired-end reads were trimmed and quality controlled by Trimmomatic (version 0.36). The clean data obtained through the above quality control procedures were used to do further analysis. For the optimal assembly results, ABySS (version 2.1.5) with multiple-kmer parameters was used for genome assembly. GapCloser (version 1.12) was then used to fill up the residual local inner gaps and correct the single base polymorphism for the final assembly results. Gene models for strain HQ were obtained using the ab initio method and identified using GeneMark. The blastp module was then used to annotate all gene models against the Nr, SwissProt, KEGG, and COG databases.

### 4.5. Gene Cloning and Expression

The ZEN hydrolase-encoding gene *zenR* was amplified from the genome DNA of strain HQ using primers *zenR*-F and *zenR*-R. The purified PCR product was then inserted into pET29a using a One Step Cloning Kit, yielding plasmid pET29a-*zenR*, which was transformed into competent *E. coli* BL21 (DE3) cells and sequenced for validation. *E. coli* BL21 cells harboring pET29a-*zenR* were subsequently grown in 100 mL LB medium containing 50 μg/mL kanamycin at 37 °C to an OD_600_ value between 0.4 and 0.6. ZenR expression was induced by settling the culture at 4 °C for 20 min, followed by the addition of IPTG at a final concentration of 0.4 mM. After another incubation at 18 °C for 12 h, the induced cells were harvested by centrifugation at 8500× *g* for 10 min at 4 °C and resuspended in 10 mL binding buffer (20 mM PBS, 20 mM imidazole, 0.5 M NaCl, pH 7.4), and ultrasonically disrupted on ice for 15 min. The intact cells were eliminated by centrifugation at 10,000× *g* for 10 min at 4 °C, and the supernatant was obtained by filtration through a 0.45 μm filter membrane. Enzyme purification was conducted on an NGC Chromatography System (Hercules, CA, USA) equipped with a HisTrap HP Ni-NTA column (Cytiva, WA, USA). First, the supernatant was loaded onto the Ni-NTA column at a flow rate of 1 mL/min. The recombinant ZenR was then eluted by a series of imidazole gradients in 20 mM phosphate buffer (pH = 7.4, 0.5 M NaCl) using three column volumes per gradient. Finally, the protein eluate was dialyzed twice at 4 °C in a dialysis solution (20 mM PBS, 50 mM NaCl, pH 7.4).

The protein concentration was estimated using the BCA protein assay kit (TianGen, Beijing, China). The subunit mass of the ZenR was determined by SDS-PAGE with a 12% separation gel and the total molecular mass of ZenR was determined by non-denaturing gel electrophoresis using a kit (Share-bio, Shanghai, China).

### 4.6. Knockout of the zenR Gene

*Rhodococcus* HQ was inoculated into 100 mL of LB medium and grown at 30 °C until the OD_600_ reached 0.4 to 0.6. Then, penicillin was added to a final concentration of 30 mg/L and cultivation continued for an additional hour. The cells were collected by centrifugation at 5000× *g* for 10 min, rinsed twice with 0.5 M sorbitol solution containing 10% glycerol, and then resuspended in 1 mL of the wash solution to obtain *Rhodococcus* competent cells. Primers LF-F/R and RF-F/R were used to amplify the upstream and downstream regions of gene *zenR*, respectively, from the genomic DNA of strain HQ. The purified PCR products were subsequently inserted together into suicide plasmid pEX18Gm to construct the recombinant plasmid pEX18Gm-ΔzenR.

For electroporation, 10 μL of the recombinant plasmid pEX18Gm-ΔzenR was mixed with 100 μL of *Rhodococcus* competent cells. The mixture was transferred into a pre-chilled 0.2 cm electroporation cuvette and pulsed at 2.0 kV. Immediately after electroporation, 600 μL of LB medium was added. The cells were allowed to recover at 30 °C, 150 rpm for 4 h. The transformation mixture was spread onto LB agar plates containing 50 μg/mL gentamicin and incubated at 30 °C. Mutant colonies from the transformation plates were isolated and inoculated into LB medium. PCR was performed to amplify the *SacB* gene to confirm the presence of the recombinant plasmid in the transformants. The single-crossover mutant was inoculated into fresh LB medium, grown at 30 °C until OD_600_ reached 0.1, and then plated on LB agar containing 20% sucrose. After incubating at 30 °C for 2 d, PCR was conducted to verify that the *zenR* gene had been successfully knocked out.

### 4.7. Enzyme Activity Assay

One unit of ZenR activity (U) was defined as the amount of enzyme required to convert 1 μmol of the ZEN per min under optimal conditions. The ZEN degradation activity of 2 μL purified recombinant ZenR was assayed in 250 μL phosphate buffer (50 mM) containing 80 μg/mL ZEN at 55 °C for 4 min. The reaction was terminated by adding an equal volume of ethyl acetate.

To determine the effect of temperature and pH on the activity of recombinant ZenR, the enzyme activity was measured at temperatures ranging from 40 °C to 75 °C and in three different buffers (phosphate citrate buffer [4.0–7.0]; phosphate-buffered saline [7.0–8.0]; glycine sodium hydroxide buffer [8.0–10.0]). The effects of potential activators and inhibitors were evaluated by adding 5 mM of different types of metal ions (Fe^3+^, Fe^2+^, Zn^2+^, Ni^2+^, Mn^2+^, Co^2+^, Cu^2+^, Ca^2+^, Mg^2+^, EDTA) and varying concentrations of ethanol (5% to 25%).

To determine the thermostability of ZenR, the enzyme was pre-incubated at various temperatures for 2, 4, 6, 8, and 10 min, respectively, before its enzyme activity was measured. Kinetic parameters were measured at a concentration range of 10–300 μM under optimal conditions. The kinetic parameters *K_m_*_,*app*_ and *V_max_* of ZenR were calculated using the Michaelis–Menten model [46].

### 4.8. Sequence Analysis and Site-Directed Mutagenesis

For the phylogenetic analysis of ZenR, multiple alignments of protein sequences were performed using ClustalW (https://www.genome.jp/tools-bin/clustalw (accessed on 20 February 2023)) and a phylogenetic tree was constructed using the maximum likelihood method. Espript 3.0 was utilized to improve graphical sequence alignment [47]. Consensus Finder Web and Weblogo 3.0 were used to predict the presence of conserved amino acid sequences in ZenR [48,49]. The structural model of ZenR was constructed by CoachFold (https://colab.research.google.com/github/sokrypton/ColabFold/blob/main/AphaFold2.i-pynb (accessed on 15 April 2023)), and the 3D model of ZEN was obtained from PubChem (https://pubchem.ncbi.nlm.nih.gov/ (accessed on 15 April 2023)). Molecular docking was performed using AutoDock software (version 4.2.6) [50], and PyMOL software (version 2.5.5) was used for image processing.

Based on multiple sequence alignment, structural comparison, and molecular docking results, the catalytic site was confirmed by site-directed mutagenesis of the predicted amino acid residues. First, the gene *zenR* was inserted into the pMD18T vector and used as a template, and the predicted site on the template was mutated using the TaKaRa MutanBEST kit (TaKaRa Co., Ltd., Dalian, China). The mutant plasmid was then transformed into competent *E. coli* DH5α for enrichment. Subsequently, the mutated *zenR* was amplified by PCR, and the gene was integrated into the pET29a vector using a one-step cloning kit. Finally, the expression plasmid was transformed into competent *E. coli* BL21 (DE3). The expression and activity assays of the various mutant enzymes were performed as described above.

### 4.9. Construction and Application of Recombinant Bacillus Subtilis

The *nprE* signal peptide was amplified from *Bacillus subtilis* 168 chromosome using nprE-F/R primers. The zenR gene was amplified from strain HQ chromosome using zenR-Bac-F/R primers. The PCR products of *nprE* and *zenR* were then assembled with BamH I-digested pAX01 vector using a one-step cloning kit to generate the recombinant plasmid. This plasmid was transformed into chemically competent *E. coli* DH5α cells for amplification. Subsequently, the recombinant plasmid was transformed into competent *B. subtilis* WB800 cells, and transformants were selected on LB agar plates containing 5 μg/mL erythromycin to obtain the recombinant *B. subtilis*.

For the production of the recombinant ZenR enzyme in the fermentation broth, an overnight seed culture of recombinant *B. subtilis* WB800 was prepared by inoculating 20 mL of LB medium supplemented with 5 μg/mL erythromycin. The seed culture was then used to inoculate 100 mL of fresh LB medium at a 2% (*v*/*v*) inoculum. The culture was grown at 37 °C, 180 rpm for 3 h and then induced to produce the recombinant ZenR enzyme by the addition of a final concentration of 1% xylose.

To simulate the application of ZenR in production, a small-scale fermentation of ZEN-contaminated cornmeal was conducted using Zhao’s method [51], in which the cornmeal underwent three stages of processing: pulping, liquefaction, and simultaneous saccharification and fermentation. Subsequently, 10 mL of fermentation broth was added to the samples after different treatment stages, respectively. The mixture was then incubated at 30 °C, 180 rpm for 3 h, followed by drying at 60 °C. To extract ZEN, 4 mL of 80% acetonitrile was added to each dried sample. To clean up the extract, 1.5 g of QuEChERS Extraction Salt (Agilent, Santa Clara, CA, USA) was added to remove impurities from the solution. A 2 mL aliquot of the supernatant was transferred to a 4 mL centrifuge tube, dried under nitrogen stream, and redissolved in 1 mL methanol for ZEN detection by HPLC.

### 4.10. Detection of ZEN and ZEN Degradation Products

The amount of ZEN was determined by high-performance liquid chromatography (HPLC, Waters e2695, Waters Corp., Milford, MA, USA) with an ultraviolet (UV) detector at 236 nm. An Eclipse XDB-C18 column (ODS, 4.6 mm × 250 mm, Zorbax) was used for chromatographic separation, and the elution was water-methanol (20:80, *v*/*v*) at 0.8 mL/min. The ZEN was eluted at 6.5 ± 0.2 min. The quantification of ZEN was accomplished by the external standard method and all the experimental data were presented as the means ± SD of three replicates. The ZEN degradation rate was calculated according to the following formula: *D_e_* = (1 − *C_t_*/*C_o_*) × 100%

*D_e_*: Biodegradation rate of ZEN; *C_o_* and *C_t_*: ZEN concentrations in control groups and experimental treatments, respectively.

To identify the ZEN degradation products, strain HQ was incubated in 10 mL MSM containing 15 μg/mL ZEN, and samples were collected at regular intervals. For mass spectral analysis, the degradation products were lyophilized and dissolved in 2 mL of acetonitrile before being filtered through a 0.22 mm filter. The LC-TOF-MS/MS analysis was operated with an AB SCIEX Triple-TOF^TM^ 5600 system (Ontario, Canada). TOF-MS/MS scan was acquired with a collision energy (CE) of 40 eV over the mass range of 100–1000 Da.

## Figures and Tables

**Figure 1 toxins-15-00688-f001:**
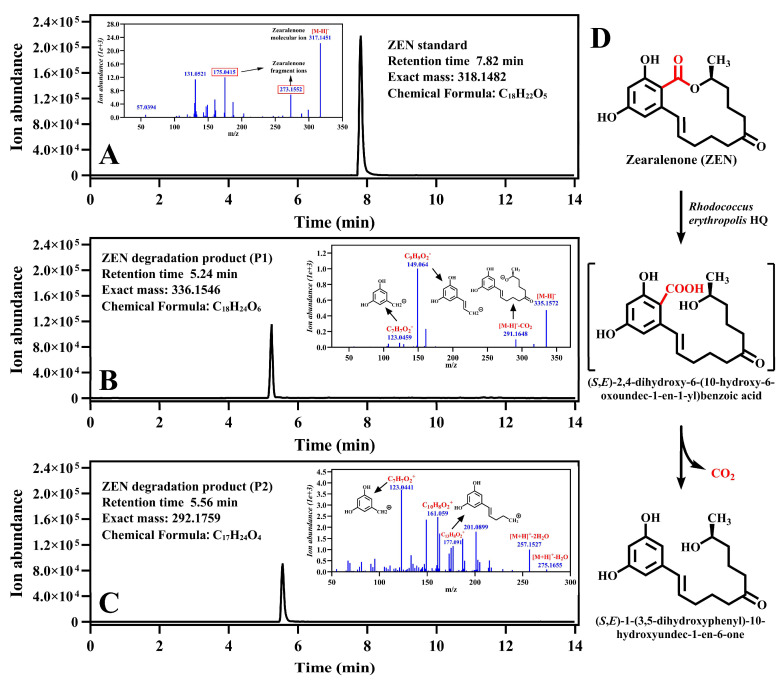
LC-MS analysis of ZEN degradation products. (**A**), extracted ion chromatogram (EIC) and mass spectrum (blue illustration) of ZEN; (**B**,**C**), EIC spectrum and mass spectrum of ZEN degradation products; (**D**), Proposed pathway for ZEN degradation by *Rhodococcus erythropolis* HQ.

**Figure 2 toxins-15-00688-f002:**
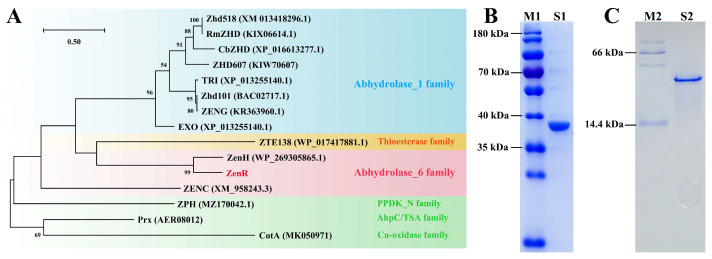
Phylogenetic tree based on the amino acid sequence of the ZEN degrading enzyme (**A**) and SDS-PAGE (**B**) and native PAGE (**C**) analysis of recombinant protein ZenR. The enzymes in the blue background belong to the α/β hydrolase_1 family, and those in the red background belong to α/β hydrolase_6 family, enzyme ZTE138 is thioesterase, enzymes in the green background are phosphotransferase (ZPH), peroxidase (Prx), and laccase (CotA), respectively. The type of enzyme is determined from the Pfam database (http://pfam.xfam.org/ (accessed on 20 February 2023)). Lane M1: protein marker (cat: 26616, Thermo Fisher); Lane S1: a subunit of recombinant protein ZenR; Lane M2: non-denaturing protein marker (cat: SKV-0052, Share-Bio); Lane S2: recombinant protein ZenR.

**Figure 3 toxins-15-00688-f003:**
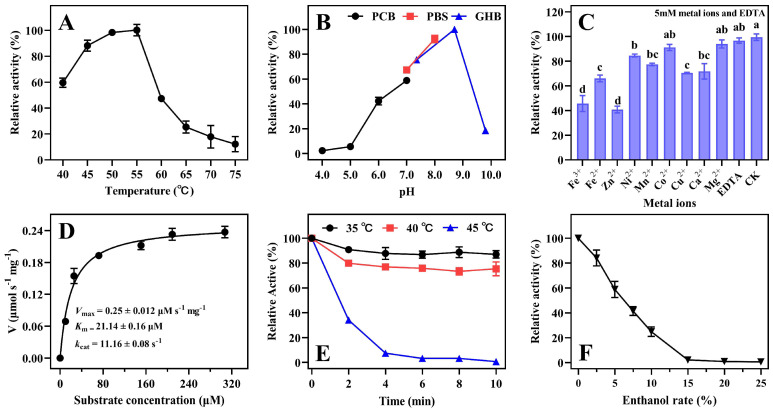
Enzymatic characterization of recombinant protein ZenR. Effects of temperature (**A**), pH value (**B**), and metal ions (**C**) on the activity of ZenR, kinetic constants for ZenR (**D**), and the effects of temperature (**E**), and organic reagents (**F**) on the stability of recombinant ZenR. The values of the relative activity are the means of three replicates and the error bars represent standard deviations. Means with different letters above the column are significantly different (*p* < 0.05). PCB: phosphate citrate buffer; PBS: phosphate-buffered saline; GHB: glycine sodium hydroxide buffer.

**Figure 4 toxins-15-00688-f004:**
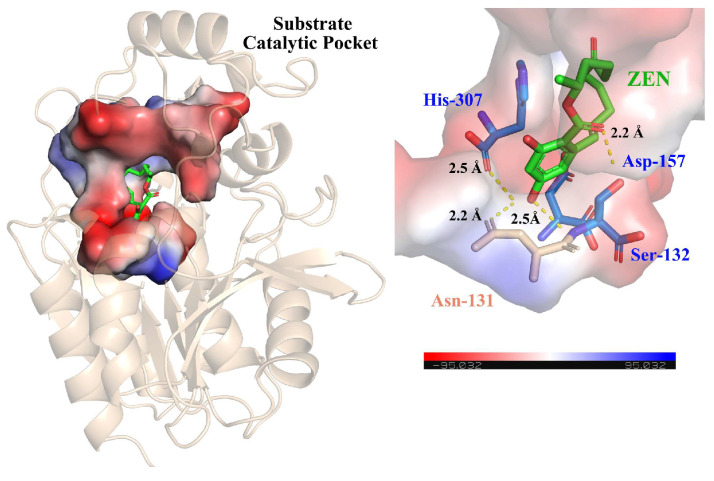
Molecular docking of the enzyme ZenR and ZEN. The left figure shows the substrate catalytic pocket of ZenR, and the right figure shows the combination of the catalytic triad and ZEN.

**Figure 5 toxins-15-00688-f005:**
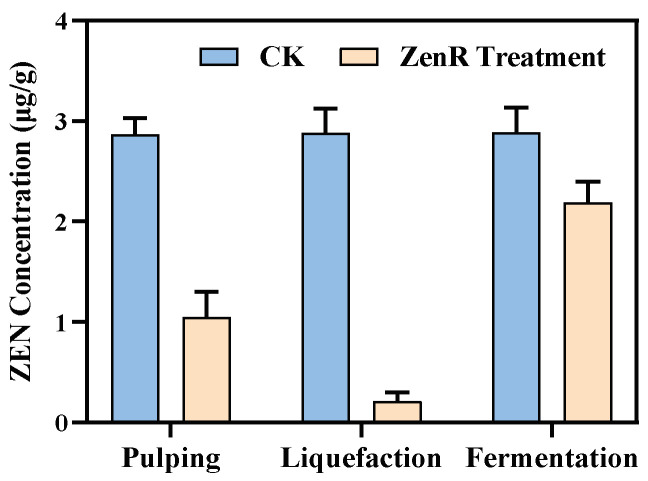
Application of recombinant *Bacillus* fermentation broth. The fermentation broth was used after the three stages of pulping, liquefaction, and simultaneous saccharification and fermentation, respectively.

## Data Availability

The data that support the findings of this study are available from the corresponding author on reasonable request.

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
