# Peer review of "A Hydrolase Produced by Rhodococcus erythropolis HQ Is Responsible for the Detoxification of Zearalenone"

_toxins, 2023, doi:10.3390/toxins15120688_

Round 1
Reviewer 1 Report
Comments and Suggestions for Authors
The manuscript “A hydrolase produced by Rhodococcus erythropolis HQ is responsible for the detoxification of zearalenone” reports the determination of a bacterial strain capable of metabolizing zearalenone (ZEA), followed by the identification of the enzyme responsible for inactivating the mycotoxin. The authors isolated a bacterial strain from the soil capable of degrading ZEA, which was identified as Rhodococcus erythropolis HQ. Under optimal degradation conditions, this strain was able to degrade 95% of the ZEA within 48 hours. Subsequently, the authors identified a hydrolase as responsible for the ZEA-degrading activity. This hydrolase, named ZenR, was successfully purified, and its enzymatic properties were determined, including its optimal temperature, pH, and enzyme activity. ZenR acts in an optimal temperature range between 50°C and 55°C, although its thermostability is greater at 40°C, when it retains approximately 80% of the enzyme activity. The structure of ZenR was identical to that of the previously described hydrolase ZenH, but the enzymatic properties of the two enzymes were distinct. The results of this study increase the number of available strains and enzymes to degrade ZEN, since ZenR may serve as a new template for the modification of ZEN hydrolases and a new resource for the industrial application of biological detoxification.
Together, this is a very interesting manuscript, which was full well conducted and presents original data concerning to the identification of a new hydrolase isolated from Rhodococcus erythropolis capable of degrading the mycotoxin zearalenone.
Therefore, we recommend the publication of this manuscript. However, we detected some linguistic errors and we believe that the whole manuscript should be corrected by a English native speaker.
Comments on the Quality of English LanguageWe detected some linguistic errors and we believe that the whole manuscript should be corrected by a English native speaker.
Author Response
Thank you very much for taking the time to review this manuscript. Please find the detailed responses below and the corresponding revisions highlighted in the re-submitted files. Please see the attachment.
Point-by-point response to Comments and Suggestions for Authors
Comments 1: Together, this is a very interesting manuscript, which was full well conducted and presents original data concerning to the identification of a new hydrolase isolated from Rhodococcus erythropolis capable of degrading the mycotoxin zearalenone. Therefore, we recommend the publication of this manuscript. However, we detected some linguistic errors and we believe that the whole manuscript should be corrected by a English native speaker.
Response 1: Thank you for your advice, this manuscript has been revised by a native English speaker and all the modifications were highlighted.
Response to Comments on the Quality of English Language
Point 1: We detected some linguistic errors and we believe that the whole manuscript should be corrected by a English native speaker.
Response 1: Thank you for your advice, this manuscript has been revised by a native English speaker and all the modifications were highlighted.

Reviewer 2 Report
Comments and Suggestions for Authors
The manuscript is well structured, balanced, and scientifically worthy. It analyzes in depth the potential role of Rhodococcus erythropolis HQ as a potential decontaminant agent of zearalenone (ZEN) in corn-based food contamined by the mycotoxigenic fungus Fusarium by using ZenR-recombinant bacteria belongs to Bacillus subtilis to detoxify corn samples from ZEN.
The authors should nevertheless improve the "Introduction" and "Conclusion" sessions by highlighting the potential role of R. erythropolis in more nosocomial contexts for human health before being used as a biological tool in the mycotoxins decontamination in food. A regulatory framework should be done and discussed in the conclusions.
Comments on the Quality of English LanguageGood English quality.
Author Response
Thank you very much for taking the time to review this manuscript. Please find the detailed responses below and the corresponding revisions highlighted in the re-submitted files. Please see the attachment.
Point-by-point response to Comments and Suggestions for Authors
Comments 1: The manuscript is well structured, balanced, and scientifically worthy. It analyzes in depth the potential role of Rhodococcus erythropolis HQ as a potential decontaminant agent of zearalenone (ZEN) in corn-based food contamined by the mycotoxigenic fungus Fusarium by using ZenR-recombinant bacteria belongs to Bacillus subtilis to detoxify corn samples from ZEN. The authors should nevertheless improve the "Introduction" and "Conclusion" sessions by highlighting the potential role of R. erythropolis in more nosocomial contexts for human health before being used as a biological tool in the mycotoxins decontamination in food. A regulatory framework should be done and discussed in the conclusions.
Response 1: Thank you for pointing this out. We have improved the Introduction section and rewritten the Conclusion section. Since the safety of Rhodococcus is unknown, we do not plan to use Rhodococcus directly as a detoxification agent, but intend to use ZenR enzyme instead, of which the biodegradation mechanism is clear.
Response to Comments on the Quality of English Language
Point 1: Good English quality.
Response 1: Thank you for the comments.

Reviewer 3 Report
Comments and Suggestions for Authors
Dear authors
Happy day
The paper titled: A hydrolase produced by Rhodococcus erythropolis HQ is re-2 sponsible for the detoxification of zearalenone, contains valuable data and interesting result. But it did not reach the optimal quality and can be improved significantly to by an art in this filed.
Kindly, put yourself in the side of the reader and re-write the parts that need more information.
1- Avoid describing material and methods in the result and the discussion parts.
2- Give the site directed mutagenesis more space in the result and discussion part.
3- No need to add your own opinion or the opinion of other researchers like "which has slowed the progress of enzyme modification" line "50".
4- You highlight some issue more than other one. For example, you neglect to mention the "16rRNA" in most of the text where there is a real need for describing it.
5- You have described some weak point in this study like it did not include some important analysis. That is ok. But you need to highlight other weak points like that the main structure of the investigated compound did not fully degrade even might be modified. Modification could be enough to cure its activity but degradation as a word is more than what you gain. So I suggest use both especially, during the describing of the data obtained from the LC-MS.
In general I like this paper and I want you to improve it.
With my pleasure

Author Response
Thank you very much for taking the time to review this manuscript. Please find the detailed responses below and the corresponding revisions highlighted in the re-submitted files. Please see the attachment.
Point-by-point response to Comments and Suggestions for Authors
Comments 1: Avoid describing materials and methods in the result and the discussion parts.
Response 1: Done as suggested. We moved materials and methods contents from the result and the discussion section to materials and methods section. See lines 373-376.
Comments 2: Give the site directed mutagenesis more space in the result and discussion part.
Response 2: Done as suggested. Please see lines 253-271.
Comments 3: No need to add your own opinion or the opinion of other researchers like "which has slowed the progress of enzyme modification" line "50".
Response 3: Done as suggested. We deleted this part.
Comments 4: You highlight some issue more than other one. For example, you neglect to mention the "16r RNA" in most of the text where there is a real need for describing it.
Response 4: Done as suggested. We added the "16r RNA" in appropriate places.
Comments 5: You have described some weak point in this study like it did not include some important analysis. That is ok. But you need to highlight other weak points like that the main structure of the investigated compound did not fully degrade even might be modified. Modification could be enough to cure its activity but degradation as a word is more than what you gain. So, I suggest use both especially, during the describing of the data obtained from the LC-MS.
Response 5: Within the context of biodegradation, the term "modification" typically denotes the formation of masked mycotoxins through original toxins and other molecules. The molecular structure remains relatively intact, and original toxins may be released during the digestive process, posing potential safety risks. In contrast, ZenR has the ability to breakdown the structure of ZEN irreversibly, ensuring comprehensive detoxification. The mechanism of action of ZenR differs significantly from modification.
Response to Comments on the Quality of English Language
Point 1: English language fine. No issues detected.
Response 1: Thank you for the comments.
